# Disharmonic Inflammatory Signatures in COVID-19: Augmented Neutrophils’ but Impaired Monocytes’ and Dendritic Cells’ Responsiveness

**DOI:** 10.3390/cells9102206

**Published:** 2020-09-29

**Authors:** Zuzana Parackova, Irena Zentsova, Marketa Bloomfield, Petra Vrabcova, Jitka Smetanova, Adam Klocperk, Grigorij Mesežnikov, Luis Fernando Casas Mendez, Tomas Vymazal, Anna Sediva

**Affiliations:** 1Department of Immunology, 2nd Faculty of Medicine, Charles University in Prague and University Hospital in Motol, 15006 Prague, Czech Republic; irena.zentsova@fnmotol.cz (I.Z.); marketa.bloomfield@fnmotol.cz (M.B.); petra.vrabcova@fnmotol.cz (P.V.); jitka.smetanova@fnmotol.cz (J.S.); adam.klocperk@fnmotol.cz (A.K.); anna.sediva@fnmotol.cz (A.S.); 2Department of Pediatrics, 1st Faculty of Medicine, Charles University in Prague and Thomayer’s Hospital, 15006 Prague, Czech Republic; 3Department of Infectious Diseases, University Hospital in Motol, 15006 Prague, Czech Republic; grigorij.meseznikov@fnmotol.cz; 4Department of Pneumology, 2nd Faculty of Medicine, Charles University in Prague and University Hospital in Motol, 15006 Prague, Czech Republic; Luis.Mendez@fnmotol.cz; 5Department of Anesthesiology and Intensive Care Medicine, 2nd Faculty of Medicine, Charles University in Prague and University Hospital in Motol, 15006 Prague, Czech Republic; tomas.vymazal@fnmotol.cz

**Keywords:** COVID-19, SARS-CoV-2, neutrophils, monocytes, dendritic cells, IFN alpha, PD-L1, innate immunity, cytokine storm, degranulation

## Abstract

COVID-19, caused by SARS-CoV-2 virus, emerged as a pandemic disease posing a severe threat to global health. To date, sporadic studies have demonstrated that innate immune mechanisms, specifically neutrophilia, NETosis, and neutrophil-associated cytokine responses, are involved in COVID-19 pathogenesis; however, our understanding of the exact nature of this aspect of host–pathogen interaction is limited. Here, we present a detailed dissection of the features and functional profiles of neutrophils, dendritic cells, and monocytes in COVID-19. We portray the crucial role of neutrophils as drivers of hyperinflammation associated with COVID-19 disease via the shift towards their immature forms, enhanced degranulation, cytokine production, and augmented interferon responses. We demonstrate the impaired functionality of COVID-19 dendritic cells and monocytes, particularly their low expression of maturation markers, increased PD-L1 levels, and their inability to upregulate phenotype upon stimulation. In summary, our work highlights important data that prompt further research, as therapeutic targeting of neutrophils and their associated products may hold the potential to reduce the severity of COVID-19.

## 1. Introduction

The COVID-19 outbreak caused by severe acute respiratory syndrome coronavirus 2 (SARS-CoV-2) has inflicted at the time of writing over 800 thousand deaths worldwide since December 2019. It is the third most serious coronavirus infection that has caused acute respiratory illness in humans since the beginning of this century, having been preceded by severe acute respiratory syndrome (SARS) in 2002 and Middle East respiratory syndrome (MERS) in 2012 [1]. Although it mainly manifests as mild respiratory and flu-like symptoms, it can progress to acute respiratory distress syndrome, sepsis, and even multiorgan failure in some individuals [2,3,4].

Apart from its direct virulence, SARS-CoV-2 may exert tissue damage via the induction of a robust inflammatory response termed hypercytokinemia or cytokine storm [2,5]. A similar phenomenon was shown to accompany severe cases of other viral infections, such as SARS, MERS, or influenza [6]. While the immunopathogenesis of such out-of-control proinflammatory reactiveness remains poorly understood, several mechanisms have been proposed. These primarily concern impaired viral clearance due to the pathogen or host genetic background, the latter resulting in aberrant NK and CD8+ cytotoxicity, downregulation of type I interferons, increased pyroptosis, or NETosis [7,8].

Barnes et al. proposed that aberrant activation of neutrophils underlies the exacerbated host response in COVID-19 [8]. Increased neutrophil count and elevated neutrophil-to-lymphocyte ratio are early indicators of SARS-CoV-2 infection, predicting severe respiratory disease, more robust cytokine response, and worse outcome [3]. Lung autopsies revealed the presence of neutrophils in lung capillaries and their extravasation into alveolar space [8]. Furthermore, transcriptomic analysis of cells from the bronchoalveolar lavage fluid of COVID-19 patients demonstrated the presence of activated neutrophils, which was markedly pronounced in COVID-19 compared to other viral pneumonias [9]. Previous studies of neutrophils in the blood of COVID-19 patients described them as dysfunctional mature cells or with immature phenotype [10,11]. And lastly, Zuo et al. revealed increased levels of neutrophil extracellular traps components in COVID-19 patients and implied their role in augmented cytokine release [12].

Decreased HLA-DR expression on COVID-19 monocytes, which could impair antigen presentation to naive T cells, was repeatedly described [10,11,13,14]. Such HLA-DR downregulation was also shown to immediately precede progression to severe respiratory failure [15]. On the other hand, monocytes are important players in inflammation, which is an integral part of COVID-19 [16].

There are only scarce records on the role of dendritic cells (DCs) in COVID-19 [14,17]. In previous years, DCs of SARS patients were found to be permissive to the virus; however, the infection failed to induce their apoptosis or maturation. The infected DCs showed low expression of antiviral interferons and only moderate upregulation of proinflammatory cytokines (TNFα and IL-6), despite significant upregulation of inflammatory chemokines (MIP-1α, RANTES, IP-10, and MCP-1) [18]. In COVID-19, single-cell RNA-sequencing analysis of peripheral blood mononuclear cells (PBMCs) revealed depletion of plasmacytoid DC (pDCs) in the blood of patients with more severe disease [13]. Activated DCs were shown to infiltrate the lungs of SARS-CoV-2 infected patients [9]. In contrast to SARS, these infiltrating cells displayed a robust IFN response, hallmarked by the expression of numerous interferon-stimulated genes (ISGs); however, no significant upregulation of IFNs was observed [9].

Innate immune mechanisms are crucial first-line antiviral defenses. To date, the research of immunopathology in COVID-19 has, however, mainly focused on the T, B, and NK cell-mediated immune responses, interferon signature dysregulation, and cytokine disbalance, with less attention to neutrophils and DCs. Herein, we describe multiple features of COVID-19 patients’ neutrophils, monocytes, and DCs. We believe that our observations add context to the general understanding of the COVID-19 immunopathology spectrum.

## 2. Materials and Methods

### 2.1. Patient Cohort and Study Design

All patients included in this study were admitted to the University Hospital in Motol, Prague, Czech Republic, between March and May 2020 and tested positive for the presence of SARS-CoV-2 RNA in nasopharyngeal swabs using reverse real time polymerase chain reaction (rtPCR). The median age of the 19 COVID-19 patients (10 female) was 73.6 ± 29.2 (range 23.6–96.7). The median age of the cohort of the 28 healthy donors (16 female) was 38.5 ± 10.4 (range 21.4–62.8). Patients were retrospectively divided into subcohorts based on severity of disease course as follows: patients with moderate course of the disease had clinical signs of pneumonia (cough, auscultation) and verified infiltration on chest X-ray or computed tomography; patients with severe course of the disease required mechanical ventilation; patients with mild course of the disease did not fulfil any of the criteria above but had a positive SARS-CoV-2 nasopharyngeal swab rtPCR; and patients with fatal course of the disease died during the course of the study. More details are in Table 1. This study was carried out in accordance with the recommendations of the Ethical Committee of the second Faculty of Medicine, Charles University in Prague, and University Hospital in Motol, Czech Republic. The protocol was approved by the Ethical Committee (EK-1225/20). All subjects gave written informed consent in accordance with the Declaration of Helsinki.

### 2.2. Media and Reagents

Cells were cultured in RPMI 1640 (Invitrogen, Carlsbad, CA, USA) medium supplemented with 10% FBS, 1% penicillin, and 1% GlutaMAX (Thermo Fisher Scientific, Waltham, MA, USA). Cells were stimulated with 1 µg/mL LPS (Sigma-Aldrich, St. Louis, MO, USA), 10 µg/mL ssRNA, 1 µg/mL R848, 50 µg/mL polyI:C (Invivogen, San Diego, CA, USA), 1 µg/mL IFNα, and 1 µg/mL IFNγ (Abcam, Cambridge, MA, USA). Various ELISAs were used for the detection of serum levels of myeloperoxidase (MPO), neutrophil elastase (NE), and IFNα (Abcam). Serum levels of G-CSF and IL-8 were evaluated by using a multiplex Luminex bead-based assay (R&D Systems, Minneapolis, MN, USA).

### 2.3. Isolation of Peripheral Neutrophils and PBMCs

Peripheral blood was collected into EDTA-coated tubes. First, peripheral blood mononuclear cells (PBMCs) were isolated using Ficoll-Paque (GE Healthcare BioSciences, Uppsala, Sweden). Neutrophils (polymorphonuclear leukocytes, PMNs) were further isolated using the Dextran sedimentation method; the remaining red blood cells were hypotonically lysed and granulocytes were washed twice in PBS without EDTA. Neutrophil purity was <90%; the major contaminants were eosinophils.

### 2.4. Neutrophil Phenotype

Peripheral blood was stained with a mixture of antibodies containing anti-lineage specific markers (CD3 clone MEM-57, CD19 clone LT19, CD20 clone LT20, CD56 clone MEM-188, CCR3 clone 5E8)-FITC, CD10-PEDY594 (clone MEM-78) (Exbio, Prague, Czech Republic), CD14-APC (clone HDC14), CD66b-PC7 (clone G10F5), CD62L-BV650 (clone DREG-56), CD11b-BV510 (clone ICRF44), CD15-A700 (clone W6D3), CD33-BV421 (clone P67.6), PDL1-PE (clone 29E.2A3) (Biolegend, San Diego, CA, USA), and HLA-DR-PerCP (clone 243) (BD Biosciences, San Jose, CA, USA) for 15 min and then, hypotonically lysed. Samples were acquired on Fortessa and analyzed using FlowJo software, version 10.5.3.

### 2.5. Monocytes and DC Phenotype

Monocytes and dendritic cells from peripheral blood were stained with the following antibodies: Lin-FITC (CD3, CD19, CD20 and CD56), CD16-A700 (clone 3G8), CD11c-APC (clone BU15), CD14-PE-DyLight594 (clone MEM-15), CD86-PE (clone BU63), CD80-A700 (clone MEM-233) (Exbio), HLA-DR-PerCP (clone L243) (BD Biosciences), PD-L1-BV510 (clone 29E.2A3), CD1c-BV510 (cloneL161), CD141-BV421 (cloneM80), and CD123-PE-Cy7 (clone 6H6) (Biolegend). FMO (fluorescence minus one) controls were used to determine the positive signal. Samples were acquired on Fortessa and analyzed using FlowJo software.

### 2.6. Cytokine Production

Neutrophils or PBMC at concentration 10^6^/mL were stimulated with LPS, ssRNA, R848, or polyI:C overnight and the cytokine levels in the culture supernatants were determined by multiplex Luminex cytokine bead-based immunoassay (R&D Systems).

For intracellular cytokine detection, 100 uL of peripheral blood from EDTA-coated tubes was stained against lineage-specific markers (CD3, CD19, CD20, CD56) conjugated with FITC, CD16-A700, CD11c-APC, CD14-PE-DyLight594 (Exbio), HLA-DR-PerCP (BD Biosciences), and CD123-PE-Cy7 (BioLegend). After RBC were lysed with BD Lysing solution (BD Biosciences), cells were fixed and permeabilized using a FixPerm kit (Thermo Fisher Scientific). Cytokines were stained using anti IL-6-PE (clone MQ2-13A5) (Biolegend), IL-1β-PE (clone CRM56) (Thermo Fisher Scientific), and TNFα-PE (clone MAb11) (Exbio), respectively. FMO controls were used to determine the cells producing the analyzed cytokines.

### 2.7. β-Galactosidase Activity

β-galactosidase activity was assessed by detecting its cleaved substrate C12FDG using flow cytometry according to manufacturer’s instructions (Thermo Fisher Scientific).

### 2.8. RT-PCR

Basal mRNA expression was analyzed in neutrophils or PBMC after isolation. Interferon-induced genes were analyzed upon IFNα or IFNγ stimulation of neutrophils for 4 h. RNA isolation, reverse transcription and RT-PCR were performed according to a previously published protocol [19]. TaqMan primer/probe sets were used (Thermo Fisher Scientific). The sample data were matched to a standard curve generated by amplifying serially diluted products using the same PCR and normalized to *GAPDH* (Thermo Fisher) to obtain the relative expression value. Real-time assays were run on an FX96 cycler (Bio-Rad, San Diego, CA, USA).

### 2.9. Western Blot

Protein detection was performed according to the previously published protocol [20]. The membranes were incubated with the following primary antibodies: anti-phosphotyrosine (p-Tyr-1000) MultiMab (#8954), anti-β-actin (clone D6A8), anti-phosphoSTAT3 (clone D3A7) (CellSignaling, Danvers, MA, USA), anti-phosphoSTAT1 (clone M135), anti-phosphoJNK1+2+3 (clone EPR18841-95), anti-phosphop38 (clone E229), and anti-phosphoIRF3 (EPR2346) (Abcam) overnight, followed by incubation with peroxidase-conjugated anti-rabbit or anti-mouse secondary antibodies for 2 h. The membranes were developed using SuperSignal West Femto (Thermo Fisher Scientific).

### 2.10. Statistics

The results obtained from at least four independent experiments are given as the median. Not all patients were involved in all experiments due to the limited amount of blood available per sample. Statistical analysis was performed using non-parametric one-way analysis of variance (ANOVA) with multiple comparisons Dunn’s post-test where applicable. A two-tailed paired Wilcoxon or unpaired Mann-Whitney *t*-test was also applied for data analysis using GraphPad Prism 8. Values of *p* < 0.05 (*), *p* < 0.01 (**) *p* < 0.001 (***) and *p* < 0.0001 (****) were considered statistically significant. Graphical abstract was created by BioRender.com.

## 3. Results

### 3.1. COVID-19 Neutrophils Are Unable to Upregulate HLA-DR and PD-L1

To address the potential role of neutrophils in COVID-19 pathogenesis, we first determined their phenotype in peripheral blood. Neutrophils were gated according to Forward Scatter (FSC-A) and Side Scatter area (SSC-A); CD15+ lineage (to exclude CD3+, CD19+, CD20+, CD56+, and CCR3+ cells) and surface levels of CD15, HLA-DR, PD-L1, CD62L, CD11b, and CD66b were then analyzed. We observed a statistically significant decrease in baseline CD15, HLA-DR, PD-L1, and CD62L expression on COVID-19 neutrophils compared to healthy donors, while CD66b and CD11b expressions were similar to controls (Figure 1A). There was, however, no correlation between the surface expression of individual markers and severity of clinical symptoms (Appendix A). The heat map of all assessed markers and principal component analysis (PCA) reducing the neutrophil phenotype of COVID-19 patients and healthy donors to two dimensions are shown in Figure 1B and Appendix A and show good separation between patients and controls.

Ex vivo stimulation of peripheral whole blood with lipopolysaccharide (LPS) led to a rapid increase in surface degranulation markers CD11b and CD66b, and decrease in CD62L, a lectin involved in granulocyte trafficking, on both patient and healthy donor neutrophils (Figure 1C and Appendix A). LPS stimulation induced increased PD-L1, HLA-DR, and CD15 surface expression in healthy donors, but significantly less so in COVID-19 neutrophils. As SARS-CoV-2 is a single-stranded RNA virus (ssRNA), we stimulated peripheral blood with ssRNA, a TLR8 ligand. Similarly to LPS, ssRNA stimulation resulted in CD11b and CD66b upregulation in both healthy donors and COVID-19 patients; however, CD62L was downregulated in patients only (Figure 1C). Intriguingly, exposure to ssRNA had no significant effect on CD15, HLA-DR, and PD-L1 expression on patients’ neutrophils but led to downregulation in healthy neutrophil to the levels of COVID-19 cells. These findings suggest that COVID-19 neutrophils are unable to dynamically regulate HLA-DR and PD-L1 expression to various stimuli.

### 3.2. COVID-19 Neutrophils Display Enhanced Degranulation of Primary Granules

Patient neutrophils exhibited significantly larger size (measured by the forward scatter parameter) and lower granularity (side scatter) compared to healthy donor cells (Figure 1D). Since the surface expression of CD11b, a gelatinase granule degranulation marker, and CD66b, a secondary granule marker, was similar in patients and healthy donors, we next analyzed the surface expression of CD63, a molecule present in primary granules, thus assessing primary granule degranulation (Figure 1E). In COVID-19 patients, an expansion of CD63+ neutrophils was apparent, and this increase was more profound in patients with severe disease. Furthermore, we evaluated serum levels of conventional primary granule components, i.e., myeloperoxidase (MPO) and neutrophil elastase (NE), and found them to be increased in COVID-19 serum. We thus demonstrate enhanced degranulation of COVID-19 neutrophils (Figure 1F).

### 3.3. Expansion of CD10-Immature Neutrophils in COVID-19 Patients

Size and granularity differ between neutrophil developmental stages [21]. We thus utilized several methods to assess neutrophil ontogenesis in COVID-19 patients. First, we estimated the proportion of immature neutrophils population utilizing surface expression of CD10, a marker discriminating mature from immature neutrophils. In the patient group, we observed significant expansion of immature CD10-neutrophils (Figure 1G). COVID-19 patients with the most severe clinical symptoms displayed significantly higher numbers of immature neutrophils (Figure 1G), which decreased over the course of the disease (Figure 1G). This cell population also exhibited a decreased surface expression of HLA-DR and PD-L1 (Appendix A). Conversely, we estimated the activity of neutrophil β-galactosidase, a marker of senescent cells, and found it to be decreased, implying a reduction in aged neutrophils in COVID-19 patients (Figure 1H). β-galactosidase activity was markedly lower in patients’ samples collected at the onset of the disease compared to samples collected 1–4 weeks later (Figure 1H). Moreover, aged neutrophils may also be distinguished by expression of chemokine receptors; while CXCR2 is downregulated, CXCR4 is enhanced, allowing the cells to return back to the bone marrow where they are depleted [22]. Analyzing the transcriptional pattern of patient neutrophils, we find *CXCR4* mRNA to be decreased and *CXCR2* increased. Taken together, these findings illustrate that COVID-19 neutrophils are predominantly of the immature phenotype (Figure 1I).

### 3.4. COVID-19 Neutrophils Produce IL-8 in Response to ssRNA and Display Pre-Activated Status

CXCR2 is a receptor for CXCL2 and IL-8 [23], which act as strong neutrophil chemoattractants. In COVID-19 patients, we detected excessive serum levels of both chemokines (Figure 1J). We also evaluated in vitro IL-8 production upon ssRNA stimulation of neutrophils and peripheral blood mononuclear cells (PBMC). Patients’ neutrophils responded to ssRNA with increased IL-8 production (Figure 2A), while healthy donors did not increase IL-8 production upon stimulation. Such different reactivity to the TLR8 ligand might be due to previous exposure to the virus and/or to a pre-existing inflammatory priming of the patients’ neutrophils. For instance, priming by GM-CSF, produced by many cell types in airways during infection, enhances IL-8 secretion in response to stimulation. Moreover, in influenza infection, viral entry, uncoating, and endosomal acidification were required for cytokine induction by human neutrophils via TLR8 and TLR7 signaling [24]. Additionally, COVID-19 PBMCs produced significantly higher amounts of IL-8 in comparison to healthy PBMCs (Figure 2A) after ssRNA stimulation. G-CSF, another key cytokine for granulocyte development and functionality, was increased in COVID-19 serum (Appendix A), which corresponds with previous findings [5].

To further assess the activation status of neutrophils in COVID-19, we measured the overall tyrosine phosphorylation in neutrophil extracts by Western blot (Appendix A) and found that compared to healthy donors, COVID-19 neutrophils exhibited higher levels of overall baseline phosphorylation, suggesting enhanced basal activation of the cells. Next, we measured the individual phosphorylation of STAT3, STAT1, JNK, p38, and IRF3 proteins and found them all to be increased in COVID-19 neutrophil whole cell lysates (Appendix A); however, it should be noted that the total levels of these proteins were not evaluated.

Overall, these findings imply that COVID-19 neutrophils are prone to inflammatory bias and receive excessive activation stimulation, which includes the “self” IL-8 positive feedback loop.

### 3.5. COVID-19 Neutrophils Are Primed to Produce Proinflammatory Cytokines in Response to ssRNA

To evaluate their responsiveness to viral products, we stimulated PBMCs with ssRNA, a TLR8 ligand, with R848, a TLR7/8 ligand, and with polyI:C (dsRNA), a TLR3, melanoma differentiation-associated gene 5 (MDA5) and retinoic acid-inducible gene-I (RIG-I) ligand, which is a by-product of viral replication. We then measured the production of proinflammatory cytokines IL-6, IL-1β, and TNFα in the supernatant. Since neutrophils do not express TLR3 and only express low amounts of TLR7 [25], isolated peripheral neutrophils were stimulated only with ssRNA. Upon ssRNA and R848 stimulation, COVID-19 PBMCs produced all proinflammatory cytokines in similar quantities to healthy donors. Interestingly, exposure to polyI:C diminished the production of IL-6, IL-1β, TNFα, and IL-12 of COVID-19 PBMCs when compared to healthy PBMCs, suggesting a partial blockade of TLR3, MDA5, and RIG-I-associated signaling pathways (Appendix A), which aligns with the recently described impaired proinflammatory response of COVID-19 pDCs to polyI:C and R848 cocktail [26].

Furthermore, after ssRNA and R848 stimulation, patients’ PBMCs released high amounts of IL-10, a predominantly anti-inflammatory/pro-tolerogenic cytokine (Appendix A), implying that PBMCs might be a prolific source of IL-10, which is also elevated in the sera of COVID-19 patients [27].

Interestingly, upon ssRNA stimulation, COVID-19 neutrophils produced significantly higher amounts of IL-6 than healthy neutrophils. Only the COVID-19, but not the healthy neutrophils, were able to increase the production of IL-1β and TNFα upon ssRNA stimulation in comparison with untreated cells (Figure 2B) indicating a pro-inflammatory bias, possibly due to priming with SARS-CoV-2 or excessive cytokine/chemokine stimulation. As shown above, COVID-19, but not healthy neutrophils, also responded to ssRNA with increased IL-8 production (Figure 2A).

### 3.6. Interferon Signatures in COVID-19 Are Decreased in PBMCs but Increased in Neutrophils

Interferon response represents an important innate immune antiviral mechanism; therefore, we studied the interferon signatures in detail. Even though COVID-19 patients exhibited enhanced levels of IFNα in the serum at the beginning of the disease (Figure 2C), this increase was only transient and declined gradually with time (Figure 2D). To study the general capacity of COVID-19 PBMCs to produce IFNα, we analyzed the basal *IFNα* mRNA expression at three different time points (1–4 weeks since the admission) (Figure 2E) and found it significantly reduced compared to healthy donors. Furthermore, PBMCs derived from patients after at least 2 weeks since admission to the hospital still released significantly lower amounts of IFNα upon ssRNA, R848, and polyI:C stimulation compared to healthy PBMCs (Figure 2F). Correspondingly, the expression of interferon-induced gene *IFIT1* in patients’ PBMCs was also decreased (Figure 2G). Overall, the IFN response in patients’ PBMCs was thus shown to be impaired.

Next, we set out to investigate the neutrophils as a potential source of type I interferons. We found that COVID-19 neutrophils displayed higher basal level of *IFNα*, *IFIT1*, and *ISG15* mRNA compared to COVID-19 PBMCs, healthy donor PBMCs, and healthy donor neutrophils (Figure 2H). There was no difference in *IFNβ* expression between COVID-19 patients’ and healthy donors’ PBMCs or neutrophils. Moreover, COVID-19 neutrophils expressed higher amounts of *ISG15* and decreased levels of suppressor of cytokine signaling 3 (*SOCS3*) mRNA (Appendix A and Figure 2I,J) upon IFNα stimulation. *IFNAR* gene expression was comparable between patients’ and controls’ cells in both PBMCs and neutrophils (Appendix A).

### 3.7. Decreased HLA-DR and Increased PD-L1 Expression on COVID-19 Monocytes and Dendritic Cells

To dissect the role of monocytes and dendritic cells in COVID-19, we first evaluated their phenotype (Figure 3A). To date, low expression of HLA-DR has only been reported on monocytes [10,11,14,28]. In our experiments, all cell types (i.e., monocytes, myeloid DCs, and pDCs) had significantly reduced HLA-DR, and increased surface PD-L1 expression. Additionally, DCs also showed decreased expression of the maturation marker CD86. Patients with severe disease displayed the highest expression of PD-L1 on both monocytes and DCs, while the expression of the maturation marker CD80 had the opposite trend (Figure 3B). Higher expression of the suppressive marker PD-L1 and low expression of maturation markers CD86 and CD80 might be a result of viral infection [29]. Re-evaluating monocyte expression of HLA-DR and PD-L1 at 1–4 weeks after admission to the hospital, we observed a tendency of these markers to recover in time; however, they did not assume the healthy controls level (Figure 3C).

Focusing on monocytes and DC subsets, we observed a mild increase in classical monocytes subset (CD14+ CD16−) (Figure 3D) and a slight decrease in non-classical monocytes (CD14− CD16+) (Figure 3D), which have been described previously [10,11]. In fact, Silvin et al. suggested monitoring this monocyte subpopulation as a fast and simple prediction tool to identify patients at risk of progression to severe disease. In alignment also with previous studies, we observed a decrease in pDCs and a shift to CD1c+ subsets in the mDC compartment in our cohort [17].

Next, we analyzed the capacity of monocytes and dendritic cells to respond to ssRNA, R848, polyI:C, and LPS. The baseline level of PD-L1 expression on COVID-19 monocytes and mDCs was increased compared to controls and the cells were unable to upregulate CD86, HLA-DR, and PD-L1 upon stimulation (Figure 3E,F). In healthy monocytes and mDCs, PD-L1 expression after ssRNA stimulation reached the unstimulated COVID-19 levels (Figure 3F). pDCs reacted only to R848 stimulation by upregulating CD86. Lastly, we examined the baseline production of proinflammatory cytokines TNFα, IL-1β, and IL-6 (Appendix A), which was enhanced compared to healthy controls.

## 4. Discussion

Although our understanding of the host immune responses to the novel SARS-CoV-2 virus is increasing by leaps, thanks to the momentum gained by global research efforts, it is still limited, particularly in the field of innate immune mechanisms. Earlier studies demonstrated major neutrophilia and enhanced levels of neutrophil-associated chemokines (IL-8, CXCL2, or G-CSF) [3,30], and pointed towards the involvement of neutrophils via the mechanisms of NETosis and necroptosis [8,12,31,32]. The most recent data from transcriptomic analyses also suggest that monocytes, T, NK, and dendritic cells are likely not the principal drivers of cytokine-mediated hyperinflammation [9,13,33] and therefore, neutrophils gradually profile as the major culprits. Here, we describe multiple features of COVID-19 patient neutrophil hyperresponsiveness and demonstrate a “stunned” state of monocytes and dendritic cells.

We show that several neutrophil properties, such as phenotype, degranulation, responses to stimuli, and cytokine production are altered in COVID-19 patients. The proinflammatory environment drives the recruitment of immature CD10-neutrophil forms from the bone marrow, especially in patients with severe disease. CD10-immature neutrophils are known to possess T cell immunostimulatory properties and prolonged survival. Of note, this population was previously noted to be expanded in G-CSF-treated donors [21] and this growth factor was also found to be increased in COVID-19 patient sera.

Neutrophils, together with mononuclear cells, are the first cells attracted to the SARS-CoV-2-infected alveoli recruited by interferons, CCL2, IL-6, IL-1β, and other cytokines [34]. On site, they eradicate the virus-infected cells, produce proinflammatory mediators, and secrete various proteases (via NETosis or independently on NETs). If this process becomes inadequately exaggerated, a “proteolytic storm” may arise and contribute to the development of acute respiratory distress syndrome (ARDS). The proteolytic storm advances the well-known cytokine storm and is due to disbalanced neutrophil serine cascade activator proteases and their inhibitors. The resulting overwhelming necroinflammation causes endotheliopathy, hypercoagulability, and diffused micro/macrothrombi formation [35]. In our experiments, we show increased levels of serum myeloperoxidase and neutrophil elastase, likely due to enhanced degranulation of COVID-19 peripheral blood neutrophils. The proteolytic storm is likely aggravated by the depletion of protease inhibitors consumed during increased NETosis [12] and by the oversecreted serine proteases. In fact, this pathological loop has already been suggested in Kawasaki-like disease, a childhood systemic vasculitis which shares several features with the recently defined pediatric multisystemic inflammatory syndrome associated with COVID-19 [36,37].

In contrast to the efflux and pro-inflammatory priming of neutrophils, a recent study by Wilk et al. described a reduction in circulating monocytes and a lack of upregulation of their proinflammatory cytokines in COVID-19, indicating their limited contribution to the COVID-19-associated cytokine storm [13]. This aligns well with our observations of downregulated HLA-DR on monocytes and DCs, which may be associated with decreased responsiveness to stimuli and suppression of CD4+ T cell responses [13,15,38]. A significant decrease in the expression of genes involved in the antigen-presentation pathways in myeloid cells was also noted by Arunachalam et al. [26]. This work, too, demonstrated the reduction in CD86 and HLA-DR on monocytes and mDCs of COVID-19 patients, which was most pronounced in subjects with severe COVID-19 infection. In our cohort, beside the low levels of HLA-DR on monocytes and DCs, we also found higher levels of the suppressive molecule PD-L1, with only slightly higher levels of proinflammatory cytokines (IL-6, IL-1β, and TNFα) expression. The alteration of the PD-1/PD-L1 axis in chronic viral infections, such as hepatitis B or human immunodeficiency virus, has been well described [39,40]. For instance, the expression of PD-1 and PD-L1 was shown to be upregulated in monocytes and DCs in a STAT3-dependent manner in response to virus-induced production of IL-10 [41]. IL-10 is also elevated in COVID-19 patients; however, the overall role of the PD-1/PD-L1 axis in acute viral infections, and COVID-19 in particular, is less clear. Moreover, COVID-19 monocytes exhibited a similar profile as monocytes from patients with hepatitis C, in which upregulation of PD-L1 and IL-10, and downregulation of HLA-DR and CD86 were the hallmark of the infection [42]. Similarly impaired CD86 upregulation on DCs treated with cytokines was previously described in COVID-19 [17].

Several pathways are involved in DC activation and/or tolerogenicity in both malignant and non-malignant conditions. In tumors, the dual state of both the proinflammatory and immunosuppressive microenvironment is well described [43,44,45,46]. Wnt5, a member of the Wnt family signaling pathway, which is wildly expressed by tumors and immune cells, activates the NFκB pathway and induces the tolerogenic phenotype via the release of IL-10 [44]. Impaired DC functions and maturation due to Wnt5 upregulation was observed in pancreatic and colon cancer [46]. Besides tumors, Wnt5 signaling is activated in sepsis or acute respiratory distress syndrome as an initiator of the lung repair process [47]. Interestingly, in a recent COVID-19 study, Wnt5 was suggested as a biomarker of disease progression, associating the decreased proinflammatory functionality of DCs with disease severity [48]. Another signaling pathway activated via NFκB in DCs engages the proinflammatory cytokinogenesis, namely IL-1β, IL-6, and TNFα. Due to the positive feedback loop, these cytokines reinforce NFκB activation and together with inflammasome assembly, subsequently favor the inflammatory reactions [49,50]. SARS-Cov-2 virus, similar to MERS and SARS-CoV, is likely capable of activating the NFκB pathway. In fact, NFκB levels were higher in SARS-CoV-2-infected lungs and suppression of this pathway enhanced IFN-mediated antiviral immunity and improved the infection outcome [51,52]. Thus, based on ours and others’ findings, we suggest that the biological pathways induced in DCs during various stages of SARS-CoV-2 may be explored further in a search for a potentially useful biological marker of disease activity.

In contrast to monocytes and DCs, COVID-19 neutrophils expressed significantly decreased levels of PD-L1 and their stimulation with ssRNA led to elevated production of proinflammatory cytokines. Despite being generally less efficient in cytokine production, neutrophils are far more frequent in peripheral blood than monocytes or DCs, particularly so in severe COVID-19. Therefore, they may, in fact, represent crucial corroborators of the cytokine storm.

In addition to general pro-inflammatory cytokines, interferons are another group of potent mediators of antiviral response and have been proposed as a treatment modality for SARS-CoV-2 infection [53]. To evade cellular immune defenses, viruses, including coronaviruses, counteract the production of type I interferons at multiple steps; for example, by binding of their proteins to dsRNA, thus blocking interactions with pathogen recognition receptors, or by abrogating TRAF3-TANK-TBK1 signaling [54]. Similar mechanisms might be involved in the impaired response to dsRNA observed in our study. On the other hand, interferons may also induce certain pro-viral events, for instance the suppression of CD4 T cells [55], thus impairing viral clearance and establishment of effective immune memory. In SARS-CoV infection, the upregulation of IFNα and IFN-stimulated genes (ISGs) was detected only in early stages of the disease and decreased over time [56], correlating with our own observations in SARS-CoV-2 patients. PBMCs displayed a generally low ex vivo and induced production of IFNα, although, surprisingly, their expression of ISG, particularly *IFIT1*, was high, in accordance with a study describing elevated expression of ISGs in bronchoalveolar lavage from COVID-19 patients [9]. Similar discrepancies were also recently pointed out in another study [26], where authors observed that ISGs are upregulated in monocytes and DCs, while pDC failed to produce IFNα and β. Despite high ISG expression by PBMCs, the source of IFNα or IFNβ was not found amongst various immune cell types (T cells, NK cells, monocytes, DCs), indicating that the neutrophils may be the likely source of the interferons.

In our study, we uncovered that neutrophils, which are also capable of producing IFNα upon selective triggering [57,58], secreted substantial amounts of IFNα, and thus, given their population size in peripheral blood, we conclude that neutrophils may substantially contribute to IFNα production in COVID-19.

Beyond the proinflammatory cytokines, also the products of lipid metabolism, namely arachidonic acid (AA) derivates—prostaglandins, leukotrienes, and thromboxane—act as proinflammatory mediators in COVID-19 and AA-deficient patients are more susceptible to a severe course of SARS-CoV-2 infection [59]. However, more studies are needed to elucidate the involvement of these pathways since inhibiting AA mediators brought conflicting results and other studies have questioned the role of AA inhibitors, such as nonsteroidal anti-inflammatory drugs (NSAID) and ibuprofen, in the worsening of COVID-19 [60].

We acknowledge that this study is not without limitations. The sample size is limited and the stratification into even smaller subcohorts is the study’s greatest weakness. In addition, not all patients were involved in all experiments due to the limited amount of blood available per sampling and the study cohort was highly heterogeneous in age, co-morbidities, and COVID-19-related risk factors. Our experiments were performed on peripheral blood cells, yet many of the innate immune processes may be specific to particular organ microenvironments. It should be emphasized that it is not clear from ours or other studies whether the aberrant functional profile of the innate immune cells is a driver of disease severity or rather a consequence of the inflammation caused by SARS-CoV-2. However, to our knowledge, this is the first study utilizing functional tests to elucidate the role of neutrophils, DCs, and monocytes in COVID-19 and as such, it provides a background for future research.

In summary, our work demonstrates the pivotal role of neutrophils in the immune disbalance in severe COVID-19 patients and supports the notion that neutrophils, but not DCs or monocytes, act as its principal drivers. It provides rationale to target neutrophils, their recruitment mediators, and neutrophil-associated products, e.g., degranulation products or NETs, as a possible therapeutic strategy in severe COVID-19 cases.

## Figures and Tables

**Figure 1 cells-09-02206-f001:**
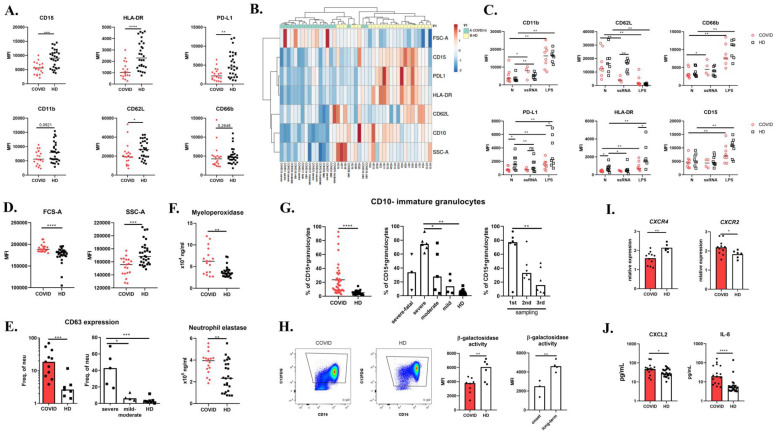
Neutrophils in COVID-19. (**A**) Peripheral blood neutrophil phenotype of 19 COVID-19 patients upon the hospital administration and 28 healthy donors (HD) was assessed by flow cytometry. (**B**) Cluster analysis of neutrophil phenotype; both rows and columns are clustered using correlation distance and average linkage. (**C**) Expression of CD11b, CD62L, PD-L1, HLA-DR, CD66b, and CD15 was analyzed in peripheral blood stimulated with 10 μg/mL ssRNA for 240 min, 1 μg/mL LPS for 30 min, or left untreated in 8 COVID-19 patients and 8 HD. (**D**) Forward Scatter (FSC-A) and Side Scatter (SSC-A) of neutrophils were analyzed in 19 COVID-19 patients and 28 HD. (**E**) CD63 expression on the peripheral neutrophil surface was analyzed in 11 COVID-19 patients and 7 HD using flow cytometry. (**F**) Serum levels of myeloperoxidase and neutrophil elastase were analyzed in 17 COVID-19 patients and 25 HD by ELISA. (**G**) Immature granulocytes were distinguished in peripheral blood by utilizing CD10 marker. CD10-CD15+CD66b+ immature neutrophils were analyzed in 19 COVID-19 patients and 28 HD by flow cytometry. The population was analyzed in three different time points during the course of the disease (1–4 weeks). (**H**) β-galactosidase activity in isolated neutrophils was assessed in 7 COVID-19 patients and 6 healthy donors by flow cytometry. (**I**) *CXCR4* and *CXCR2* expression on isolated neutrophils was analyzed by RT-PCR in 11 COVID-19 patients and 6 HD. Expression was normalized to *GAPDH.* (**J**) Serum levels of CXCL2 and IL-8 in COVID-19 patients (*n* = 17) and HD (*n* = 25) were analyzed by ELISA. Statistical analysis was performed using the Wilcoxon paired or Mann–Whitney unpaired *t*-test. Values of *p* < 0.05 (*), *p* < 0.01 (**), *p* < 0.001 (***), and *p* < 0.0001 (****) were considered statistically significant.

**Figure 2 cells-09-02206-f002:**
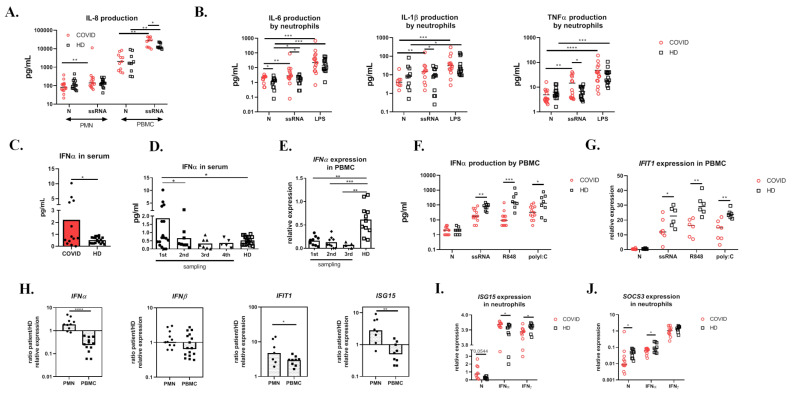
Cytokine production. (**A**) IL-8 concentration in cell-free supernatants of isolated neutrophils and PBMCs from COVID-19 (*n* = 15) and HD (*n* = 15) upon 10 μg/mL ssRNA stimulation overnight detected by Luminex. (**B**) Isolated neutrophils from 14 COVID-19 patients and 14 HD were stimulated overnight with 10 μg/mL ssRNA or left untreated and the production of IL-6, IL-1β, and TNFα was analyzed with Luminex. (**C**) Serum level of IFNα in 13 COVID-19 and 19 HD analyzed by ELISA. (**D**) Dynamics of serum levels of IFNα during the course of the disease (four different time points in 1–4 weeks). (**E**) Dynamics of *IFNα* expression by COVID-19 (*n* = 9) and HD (*n* = 11) PBMCs during the course of the disease (four different time points in 1–4 weeks) detected by RT-PCR. (**F**) IFNα production by COVID-19 (*n* = 13) and HD (*n* = 8) PBMCs upon 10 μg/mL ssRNA, 1 μg/mL R848, and 50 μg/mL polyI:C overnight stimulation detected in cell-free supernatants by Luminex. (**G**) *IFIT* expression by COVID-19 (*n* = 6) and HD (*n* = 6) PBMCs upon 10 μg/mL ssRNA, 1 μg/mL R848, and 50 μg/mL polyI:C stimulation for 4 h. Expression was normalized to *GAPDH.* (**H**) *IFNα*, *IFNβ*, *IFIT*, and *ISG15* expressions by COVID-19 PBMCs (*n* = 9–16) and neutrophils (*n* = 7–10). The relative expression was calculated using the following formula (expressed gene/*GAPDH* of COVID-19/median of (Expressed gene/*GAPDH* of all HD samples)). (**I**,**J**) *ISG15* and *SOCS3* expression by COVID-19 (*n* = 11) and HD (*n* = 8) neutrophils stimulated with IFNα (1 µg/mL), IFNγ (1 µg/mL) for 4 h, or left untreated. The expression was normalized to *GADPH.* Statistical analysis was performed using the Wilcoxon paired or Mann–Whitney unpaired *t*-test. Values of *p* < 0.05 (*), *p* < 0.01 (**), *p* < 0.001 (***), and *p* < 0.0001 (****) were considered statistically significant.

**Figure 3 cells-09-02206-f003:**
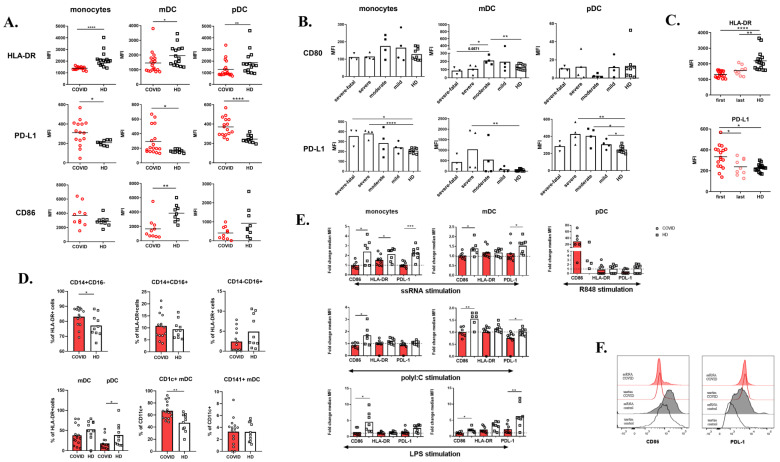
Phenotype of monocytes and dendritic cells. (**A**) HLA-DR, PD-L1, and CD86 expression on monocytes, myeloid, and plasmacytoid dendritic cells (mDCs, pDCs) detected by flow cytometry. Data are expressed as mean fluorescence intensity (MFI). (**B**) CD80 and PD-L1 expression on monocytes, mDCs, and pDCs of COVID-19 (*n* = 15) patients divided according to the severity of the disease. Data are expressed as MFI. (**C**) HLA-DR and PD-L1 expression on COVID-19 monocytes detected after admission to the hospital (“first”, *n* = 17) and after time of 2–4 weeks (“last”, *n* = 9). Data are expressed as MFI. (**D**) Peripheral blood monocytes and DC subpopulations of COVID-19 patients (*n* = 14) and HD (*n* = 10) assessed by flow cytometry. (**E**) Whole blood was stimulated with 10 μg/mL ssRNA, 50μg/mL polyI:C, 1 μg/mL R848, 1 μg/mL LPS, or left untreated overnight and then CD86, HLA-DR, and PD-L1 expression on the surface of monocytes, mDCs, and pDCs were analyzed. Data are expressed as fold change (stimulated MFI/unstimulated MFI) of nine COVID-19 patients and eight HD. (**F**) Representative histograms of CD86 and PD-L1 expression upon ssRNA stimulation of COVID-19 patients and HD. Statistical analysis was performed using the Wilcoxon paired or Mann–Whitney unpaired *t*-test. Values of *p* < 0.05 (*), *p* < 0.01 (**), *p* < 0.001 (***), and *p* < 0.0001 (****) were considered statistically significant.

**Table 1 cells-09-02206-t001:** Demographic and clinical characteristics of COVID-19 patients.

Subcohorts	N	Age	Sex	Therapy
Mild	4	30.8 ± 25.6	2F, 2M	None
Moderate	6	82.6 ± 7.4	4F, 2M	4 Hydroxychloroquin, 2 azithromycin
Severe	6	48.4 ± 18.2	3F, 3M	5 Hydroxychloroquin, 3 azithromycin
Severe-Fatal	3	83.4 ± 6.2	1F, 2M	1 azithromycin, 1 corticosteroids

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
