# Peer review of "Disharmonic Inflammatory Signatures in COVID-19: Augmented Neutrophils’ but Impaired Monocytes’ and Dendritic Cells’ Responsiveness"

_cells, 2020, doi:10.3390/cells9102206_

Round 1
Reviewer 1 Report
The manuscript is an excellent scientific work. I congratulate the authors (this is not usual in my evaluations). The methodology used is adequate. The results are very interesting and support the discussion. I only have the following comments.
I. Major Comments:
1. In the inflammatory response the activation of the transcription factor NF-kB is key. In this regard, oxidative stress is essential in the activation of NF-kB. Activation of NF-kB is the first step to increase the expression (mRNA) and plasma levels of pro-inflammatory cytokines (TNF-alpha, IL-1beta and IL-6). Include a paragraph about it:
Suggested references:
Prevention of liver ischemia reperfusion injury by a combined thyroid hormone and fish oil protocol. J Nutr Biochem. 2012; 23: 1113-20. PMID: 22137030
Docosahexaenoic acid and hydroxytyrosol co-administration fully prevents liver steatosis and related parameters in mice subjected to high-fat diet: A molecular approach. Biofactors. 2019; 45 (6): 930-943.
PMID: 31454114
Suppression of high-fat diet-induced obesity-associated liver mitochondrial dysfunction by docosahexaenoic acid and hydroxytyrosol co-administration. Dig Liver Dis. 2020; 52: 895-904.
PMID: 32620521
2. In activating the inflammatory response, I suggest discussing the role of arachidonic acid, as a precursor of eicosanoids and lipoxins directly related to inflammation.
II. Minor comments:
1. Improve the writing of the study objective.
2. In the discussion I suggest including a figure that summarizes the main results. Highlighting the molecular mechanisms involved.
Author Response
Dear reviewer,
Thank you very much for taking time to review our manuscript. We appreciate the issues raised in your comments which we tried to address thoroughly and hope you will find the answers and amendments satisfactory.
The manuscript is an excellent scientific work. I congratulate the authors (this is not usual in my evaluations). The methodology used is adequate. The results are very interesting and support the discussion. I only have the following comments.
- Major Comments:
- In the inflammatory response the activation of the transcription factor NF-kB is key. In this regard, oxidative stress is essential in the activation of NF-kB. Activation of NF-kB is the first step to increase the expression (mRNA) and plasma levels of pro-inflammatory cytokines (TNF-alpha, IL-1beta and IL-6). Include a paragraph about it:
Suggested references:
Prevention of liver ischemia reperfusion injury by a combined thyroid hormone and fish oil protocol. J Nutr Biochem. 2012; 23: 1113-20. PMID: 22137030
Docosahexaenoic acid and hydroxytyrosol co-administration fully prevents liver steatosis and related parameters in mice subjected to high-fat diet: A molecular approach. Biofactors. 2019; 45 (6): 930-943. PMID: 31454114
Suppression of high-fat diet-induced obesity-associated liver mitochondrial dysfunction by docosahexaenoic acid and hydroxytyrosol co-administration. Dig Liver Dis. 2020; 52: 895-904. PMID: 32620521
- We fully agree, the NFκB activation is, indeed, a crucial step in inflammatory responses, therefore, we added this part into the Discussion.
- In activating the inflammatory response, I suggest discussing the role of arachidonic acid, as a precursor of eicosanoids and lipoxins directly related to inflammation.
- Thank you for this suggestion, we have also added this aspect into the Discussion.
- Minor comments:
- Improve the writing of the study objective.
- We have improved the Objective.
- In the discussion I suggest including a figure that summarizes the main results. Highlighting the molecular mechanisms involved.
- Thank you very much for the suggestion. We have created a figure describing the main molecular mechanisms involved in proinflammatory response in COVID-19 patients and since it overlaps with the graphical abstract, we decided to use it as a new graphical abstract instead if the reviewer agrees.
Reviewer 2 Report
To the authors Zuzana Parackova et al. provide a balanced assessment of the status of immune dysregulation, NETosis and neutrophil-associated cytokines, CETP inhibitors in COVID-19. The article highlights important data that might have been overlooked when promulgating the clinical value of immune status and SARS-CoV2 pathophysiology and related studies.
Major point to be considered in the next versions:
1.Patient cohort and study design: strengths, and weaknesses of the study should be better highlighted and the small sample size should be emphasized.
2. ELISA tests: did the authors employ DuoSet o Quantikine tests? Did the authors check for ELISA limitations (i.e. temporary readouts biases and limited antigen information: idetection is based on enzyme/substrate reactions and therefore readout must be obtained in a short time span, information limited to amount or presence of the antigen in the sample
3. Did the authors employed unstained controls and isotype antibody controls while performing FACS analysis? Those are foundamental informationa and should be mentioned in the manuscript.
4. RT-PCR: the authors should also provide absolute count of the healthy samples/controls.
5. For all Western blot figures and tests, densitometry readings/intensity ratio of each band should be included; the whole Western blot showing all bands and molecular weight markers should be included in the Supplementary Materials.
6. The image quality deserve some beautification and higher resolution.
7. While assessing dendritic cells responsiveness, several biological pathways have been uncovered to significantly impact on DCs proficiency in both non-malignant and malignant conditions (namely WNT pathway, glutaminolysis, mitophagy, mitochondrial dynamics, OXPHOS, glycolysis). Can the authors comment on this? These biological connections might increase the translational relevance of the authors findings, by prompting future studies designed to help clinicians to optimize molecular testing and relevant clinical application for their patients.
8. A native speaker revision would improve the manuscript's quality.
Author Response
Dear reviewer,
Thank you very much for taking time to review our manuscript. We appreciate the issues raised in your comments which we tried to address thoroughly and hope you will find the answers and amendments satisfactory.
Major point to be considered in the next versions:
- Patient cohort and study design: strengths, and weaknesses of the study should be better highlighted and the small sample size should be emphasized.
- We would like to thank the reviewer for his/her valuable comments. We agree, the study is not without limitations and we have already emphasized the weaknesses at the end of the Discussion. Additionally, we now comment on the limited sample size and highlight the strengths.
- ELISA tests: did the authors employ DuoSet or Quantikine tests? Did the authors check for ELISA limitations (i.e. temporary readouts biases and limited antigen information: idetection is based on enzyme/substrate reactions and therefore readout must be obtained in a short time span, information limited to amount or presence of the antigen in the sample
- For ELISAs, we have used simple-step kits from Abcam and not from R&D Systems. We opted for these kits due to our previous good experiences with these particular products but we did not check for the specification. For cytokine detection, we utilized Luminex kits from R&D Systems (a multiple bead- based assay).
- Did the authors employed unstained controls and isotype antibody controls while performing FACS analysis? Those are foundamental informations and should be mentioned in the manuscript.
- Thank you very much. Yes, we have used FMO staining and we have added this information into the Method section in the manuscript.
- RT-PCR: the authors should also provide absolute count of the healthy samples/controls.
- We apologize for the lack of clarity. We used a standard curve generated by serial diluted cDNA to check the efficiency of the reaction, which does not provide absolute numbers of expression. In case, the reviewer meant Figure 2H, the data were expressed as ratio of patient relative expression/healthy relative gene expression. The formula is described in the figure legend.
- For all Western blot figures and tests, densitometry readings/intensity ratio of each band should be included; the whole Western blot showing all bands and molecular weight markers should be included in the Supplementary Materials.
- Thank you very much for this comment. We have determined the densitometry intensity using ImageJ software and added graphs into the figure. The outcropped gels were submitted within the Supplementary Material.
- The image quality deserves some beautification and higher resolution.
- Thank you for the comment. We have prepared figures in 600dpi and we will provide them. However, the submission system in the Cells required to include figures within the text, therefore the quality was automatically decreased.
- While assessing dendritic cells responsiveness, several biological pathways have been uncovered to significantly impact on DCs proficiency in both non-malignant and malignant conditions (namely WNT pathway, glutaminolysis, mitophagy, mitochondrial dynamics, OXPHOS, glycolysis). Can the authors comment on this? These biological connections might increase the translational relevance of the authors findings, by prompting future studies designed to help clinicians to optimize molecular testing and relevant clinical application for their patients.
- Thank you very much for this interesting suggestion. We agree, many important biological pathways that are active in malignant and non-malignant conditions have the effect on the DC reactivity. There are several articles discussing a potential role of various signalling pathways in COVID-19, such as NFkB or Wnt. We combined this reviewer’ s suggestion with the recommendation from R1 and added some thoughts about various pathways involvement in inflammation and DC reactivity to the Discussion section.
- A native speaker revision would improve the manuscript's quality.
- The manuscript was edited by a native English speaker.
Reviewer 3 Report
The authors propose a very interesting manuscript describing the crucial role of neutrophils in hyperinflammation associated with COVID19 and the dysregulated response of these cells and also of monocytes and DCs. The authors also shown a transient increase of IFNa in sera of COVID patients that declined gradually with time and was sustained by neutrophils, that also increased production of IFN regulated genes.
The proposed research has a great limitation, the low number of patients (only 19 COVID patients), also considering that they are stratified in mild, moderate, severe and fatal cases (so the differences in analyzed markers are difficult to be considered statistically significant) and (as recognized by the authors) not all of their samples were analyzed in all experiments. However, due to the urgent need to advance the knowledge on the immune response in COVID, I think that this work could be of importance and deserve to be considered for publication also to suggest further studies in this direction.
I then suggest some revisions, prior to publication:
- Table 1 legend must be better specified, as example: Demographic and clinical characteristics of COVID-19 cases.
- Results:
1) it could be more clear for the readers if COVID patients and HD would be shown with colours different from red and blue in the cluster of Fig. 1b. It would be confounding since the cluster is also colored with red and blue
2) it could be more clear for the readers if authors show results of figure 1c and suppl. figure 1c together: CD11b, CD62L, PDL1 and HLA-DR are shown both in fig 1c and suppl 1c, while Cd66b only in suppl 1c; in suppl 1c only one COVID patient is shown but no HD. In suppl. 1c CD15 is shown but it is not mentioned in the text
3) how the authors can explain that HD PMN did not increase IL8 production upon ssRNA stimulation, while HD PBMC clearly increased IL8 although in lower amounts in comparison with patients? Please discuss this point in the discussion
4) authors interestingly show the high phosphorylation state of patients’ neutrophils in Fig. suppl F. Although protein load is shown with beta actin, however levels of total STAT3, STA1, JNK, p38 and IRF3 should also be shown. Is it an increase also in the total amount of these proteins? Or only in phosphorylation state of these proteins? These data must also be discussed in the discussion section
5) page 9 lane 299: Authors say: “ Only the COVID-19, but not the healthy neutrophils, were able to produce IL-1β and TNFα (Figure 2B)” . Probably authors mean more IL1β and TNFα in comparison with untreated cells?
6) page 9 lane 317: authors say “We found that COVID-19 neutrophils displayed higher basal level of IFNα, IFIT1 and ISG15 mRNA compared to COVID-19 PBMCs, healthy donor PBMCs and healthy donor neutrophils (Figure 2H)”. However figure 2H shows only the comparison between PMN and PBMC of COVID patients
7) page 9 lane321: CXCL10 is not represented in fig 2I, J nor suppl 1F
8) IL-12p70 production in PBMC, fig suppl 2A is not commented in the text
- Discussion: interestingly a very recent work by Arunachalam and colleagues (Science) demonstrated impaired IFN production by pDCs but a transient increase of serum IFNa (as in this manuscript), supposed likely of lung origin, and showed a reduced expression of HLA-DR in myeloid cells, mainly in severely affected patients. It could be interesting to discuss the findings of the authors in view of the mentioned article.
Author Response
Dear reviewer,
Thank you very much for taking time to review our manuscript. We appreciate the issues raised in your comments which we tried to address thoroughly and hope you will find the answers and amendments satisfactory.
I then suggest some revisions, prior to publication:
- Table 1 legend must be better specified, as example: Demographic and clinical characteristics of COVID-19 cases.
- Thank you very much. We have amended the Table legend.
- Results:
- it could be more clear for the readers if COVID patients and HD would be shown with colours different from red and blue in the cluster of Fig. 1b. It would be confounding since the cluster is also colored with red and blue
- We have changed the heat map parameters and distinguished the colours of COVID-19 and HD groups.
- it could be more clear for the readers if authors show results of figure 1c and suppl. figure 1c together: CD11b, CD62L, PDL1 and HLA-DR are shown both in fig 1c and suppl 1c, while Cd66b only in suppl 1c; in suppl 1c only one COVID patient is shown but no HD. In suppl. 1c CD15 is shown but it is not mentioned in the text
- Thank you very much for this suggestion. We aimed to display the data with the most striking differences between the cohorts in Fig.1 to maintain simplicity. However, we made the suggested changes to the Fig. 1 and mentioned the expression of CD15 in the main body of the manuscript. Our original intention for Suppl. Fig. 1C was only to show representative histograms of the changes and how different duration of stimulation effected the expressions. We were conscious if adding another 5 histograms from healthy control would not compromise the clarity of the figure. However, we added the missing histograms as suggested.
- how the authors can explain that HD PMN did not increase IL8 production upon ssRNA stimulation, while HD PBMC clearly increased IL8 although in lower amounts in comparison with patients? Please discuss this point in the discussion
- Thank you very much for this interesting point for the discussion. We believe that COVID PMNs might have released IL-8 due to GM-CSF priming, a cytokine elevated in COVID-19 patients, or because they already encounter the virus and PMNs require priming steps for IL-8 production. IL-8 acts as a chemokine for neutrophils and PMNs probably need a 2-step activation for producing a “selfactivating” chemokine. On the other hand, human PBMCs are known to produce higher levels of cytokines when compared to neutrophils and they probably do not need priming steps to induce the IL-8 in order to activate neutrophils. We address this point in the Result section where we mentioned IL-8 production.
- authors interestingly show the high phosphorylation state of patients’ neutrophils in Fig. suppl F. Although protein load is shown with beta actin, however levels of total STAT3, STA1, JNK, p38 and IRF3 should also be shown. Is it an increase also in the total amount of these proteins? Or only in phosphorylation state of these proteins? These data must also be discussed in the discussion section
- Although a most valid point, we did not determine the total proteins levels of the abovementioned proteins as part of the experiments. We acknowledge this shortcoming briefly in the result section. We can, however, show the densitometry readings as the reviewer 2 already asked for.
- page 9 lane 299: Authors say: “ Only the COVID-19, but not the healthy neutrophils, were able to produce IL-1β and TNFα (Figure 2B)” . Probably authors mean more IL1β and TNFα in comparison with untreated cells?
- Indeed, the reviewer is right. We corrected this.
- page 9 lane 317: authors say “We found that COVID-19 neutrophils displayed higher basal level of IFNα, IFIT1 and ISG15 mRNA compared to COVID-19 PBMCs, healthy donor PBMCs and healthy donor neutrophils (Figure 2H)”. However figure 2H shows only the comparison between PMN and PBMC of COVID patients.
- We apologize for the lack of clarity. However, the figure does not demonstrate individual subjects’ results but the ratio of the expression of the patients’ and healthy cells (the median of the values obtained from RTPCR from neutrophils or PBMCs). Figure legend “The relative expression was calculated using the following formula (expressed gene/GAPDH of COVID-19/median of (Expressed gene/GAPDH of HD samples)”.
- page 9 lane321: CXCL10 is not represented in fig 2I, J nor suppl 1F
- We would like to apologize for this mistake. During the manuscript editions, we have not noticed that old data were still included in the text. We deleted them. Thank you for notifying us.
- IL-12p70 production in PBMC, fig suppl 2A is not commented in the text
- Thank you for the comment, now we are mentioning IL-12 in the text.
- Discussion:
- interestingly a very recent work by Arunachalam and colleagues (Science) demonstrated impaired IFN production by pDCs but a transient increase of serum IFNa (as in this manuscript), supposed likely of lung origin, and showed a reduced expression of HLA-DR in myeloid cells, mainly in severely affected patients. It could be interesting to discuss the findings of the authors in view of the mentioned article.
- Thank you very much for this interesting article! Their data aligns with our nicely. We have included it in the Discussion section.
Round 2
Reviewer 1 Report
Excellent manuscript. The authors made all changes suggested.
Reviewer 3 Report
The manuscript is now acceptable for publication.
It is a good work, thanks to the authors for the corrections